# Roles of α-Synuclein and Disease-Associated Factors in *Drosophila* Models of Parkinson’s Disease

**DOI:** 10.3390/ijms23031519

**Published:** 2022-01-28

**Authors:** Mari Suzuki, Kazunori Sango, Yoshitaka Nagai

**Affiliations:** 1Diabetic Neuropathy Project, Department of Diseases and Infection, Tokyo Metropolitan Institute of Medical Science, Setagaya, Tokyo 156-8506, Japan; sango-kz@igakuken.or.jp; 2Department of Neurotherapeutics, Graduate School of Medicine, Osaka University, Suita 565-0871, Japan; 3Department of Neurology, Faculty of Medicine, Kindai University, Osaka-Sayama 589-8511, Japan

**Keywords:** *Drosophila*, α-Synuclein, Parkinson’s disease

## Abstract

α-Synuclein (αSyn) plays a major role in the pathogenesis of Parkinson’s disease (PD), which is the second most common neurodegenerative disease after Alzheimer’s disease. The accumulation of αSyn is a pathological hallmark of PD, and mutations in the *SNCA* gene encoding αSyn cause familial forms of PD. Moreover, the ectopic expression of αSyn has been demonstrated to mimic several key aspects of PD in experimental model systems. Among the various model systems, *Drosophila melanogaster* has several advantages for modeling human neurodegenerative diseases. *Drosophila* has a well-defined nervous system, and numerous tools have been established for its genetic analyses. The rapid generation cycle and short lifespan of *Drosophila* renders them suitable for high-throughput analyses. PD model flies expressing αSyn have contributed to our understanding of the roles of various disease-associated factors, including genetic and nongenetic factors, in the pathogenesis of PD. In this review, we summarize the molecular pathomechanisms revealed to date using αSyn-expressing *Drosophila* models of PD, and discuss the possibilities of using these models to demonstrate the biological significance of disease-associated factors.

## 1. Introduction

Parkinson’s disease (PD) is a progressive neurodegenerative disease characterized by motor symptoms, such as a resting tremor, bradykinesia, and rigidity, as well as non-motor symptoms, such as dementia, depression, autonomic failure, and hallucinations [1]. The progressive degeneration of dopaminergic (DA) neurons in the substantia nigra is the main cause of the motor symptoms, and dopamine replacement therapy is widely used to improve these symptoms, although it does not attenuate disease progression. Pathological hallmarks of PD are the deposition of Lewy bodies (LBs), which are mainly composed of fibrillar α-synuclein (αSyn) [2,3]. αSyn is a 140 amino acid protein that is abundantly expressed in neurons, particularly in presynaptic terminals [4]. The gene encoding αSyn, *SNCA*, was the first gene to be identified as a causative gene for familial PD [5]. To date, six missense mutations (A30P, E46K, H50Q, G51D, A53T, and A53E) and duplication and triplication mutations of *SNCA* have been found to cause familial PD [6,7,8,9,10,11,12,13]. Most importantly, genome-wide association studies (GWAS) also identified single nucleotide polymorphisms (SNPs) in the *SNCA* gene as major risk factors for sporadic PD, which comprises the majority of PD patients [14,15,16]. These findings strongly indicate that αSyn plays a crucial role in the pathogenesis of PD.

Transgenic animal models of PD have been generated following the identification of *SNCA* gene mutations that are causally linked to familial PD. The expression of wild-type (WT) or mutant αSyn has been experimentally shown to mimic several aspects of PD in various animal species, including *Caenorhabditis elegans*, *Drosophila melanogaster*, and rodents [17,18,19]. *Drosophila*, commonly known as the fruit fly, provides a simple, yet powerful, in vivo system for modeling human neurodegenerative diseases. The advantages of using *Drosophila* for studying human diseases are summarized in Table 1. *Drosophila* has a well-defined nervous system, and has homologues of approximately 75% of human disease genes [20]. Its rapid generation cycle and short lifespan enable the creation of multiple genetically modified flies, allowing for the investigation into the effects of aging-associated disease phenotypes in a timesaving manner. It should be noted that the abundant genetic information and useful genetic tools of *Drosophila* are widely and publicly shared. FlyBase is the leading database for genetic and genomic information of *Drosophila*, which is also linked to information regarding available animal stocks and reagents, such as cDNA clones and antibodies. Genome-wide mutant and RNA interference (RNAi) fly libraries, the GAL4/*upstream activating sequence* (*UAS*)-targeted transgene expression system, and genome-editing systems have been generated, and various genetically-engineered fly strains for most of genes are commercially available from public stock centers at low cost. These advantages of *Drosophila* models enable their use in genome-wide modifier screenings and high-throughput drug screenings.

## 2. Modelling PD by αSyn Expression in *Drosophila*

Feany and Bender first developed transgenic PD model flies expressing human αSyn, which recapitulate several features of human PD, including locomotor dysfunction, LB-like inclusion body formation, and the progressive loss of DA neurons [18]. Although *Drosophila* does not have a homolog of *SNCA*, the fact that pathogenic mutations and the multiplication of *SNCA* cause PD with a dominant inheritance pattern in patients implicates a toxic gain-of-function mechanism; thus, transgenic flies expressing WT or mutant αSyn were established to study the molecular pathogenesis of PD. From then on, various types of αSyn flies have been established to elucidate the effects of familial PD-linked αSyn mutations, its post-translational modifications, and the molecular mechanisms of αSyn-induced neurodegeneration. The recent development of genome analysis technologies has led to the identification of various PD-associated gene mutations and polymorphisms in PD patients. Furthermore, αSyn-expressing transgenic flies have also been used for studying how these genetic factors interact with αSyn in vivo.

Various characteristics of PD have been successfully recapitulated in *Drosophila* (Table 2). The most common behavioral analysis used for flies expressing αSyn under the control of the pan-neuron- or DA-neuron-specific GAL4 driver is the climbing assay, which utilizes the fly’s intricate negative geotaxis behavior to assess their locomotor function. When tapped to the bottom of a vial, adult flies will climb back upwards towards the top, and the number or velocity of the climbing flies is scored [21]. Semi-automated and fully automated assays have also been developed to quantitatively characterize the climbing behavior at high parametric resolution, such as by analyzing total distance, straightness, and gait pattern. For example, “iFly” is a computerized tracking system that enables the researcher to obtain three-dimensional views of individual flies [22]. Another system is “fly vertically rotating arena for locomotion” (fly-VRL) with the custom software “Fly Contour-based Tracker” (FlyConTra), which is an inexpensive fully automated assay [23].

Non-motor behaviors of PD, including abnormal sleep behavior, olfactory deficits, anxiety, and cognitive dysfunction have been evaluated in PD model flies. The *Drosophila* Activity Monitor system (TriKinetics, Waltham, MA, USA) is a widely used set of devices for recording spontaneous *Drosophila* locomotor behavior. Long-term data recording, and the ability to monitor behavior in the darkness, make this system particularly suitable for studying sleep behavior and circadian rhythms [24,25]. Olfactory deficits, anxiety-like behavior, and cognitive dysfunction of αSyn-expressing flies have also been demonstrated by the odor acuity/discrimination assay, the open-field assay, and the T-maze assay, respectively [26,27]. Shortened lifespan is commonly seen in αSyn-expressing PD flies, although the life expectancy of PD patients with an average age of onset (60 years) is now almost comparable with that of the general population, owing to substantial advances in medical care [28,29].

Histological analysis for evaluating DA neuron loss using anti-tyrosine hydrolase (TH) antibodies is widely performed. The method of assessing changes in external eye morphology using an eye-specific GAL4 driver was originally developed for modifier screening. As the eye is a non-vital organ, the effects of highly toxic genes can be assessed in adult flies without problems of lethality, and external eye observation enables a rapid readout for large-scale screens. Whereas several disease proteins, such as the Machado–Joseph disease protein with an expanded polyglutamine stretch, causes severe degeneration that is easily observed by light microscopy (LM) [30], αSyn expression causes mild degeneration that is somewhat difficult to evaluate externally by LM, and often requires scanning electron microscopy or histological analyses [18]. To save time and effort, Burr and colleagues expressed membrane-targeted green-fluorescent protein together with the αSyn protein in the compound eyes of *Drosophila*, which enabled a quantitative analysis of degeneration using optical fluorescence microscopy [31]. In addition, physiological analyses, such as electoretinography (ERG) [32] and electrophysiological recordings from projection neurons [27] or neuromuscular junctions [33], which directly assess neuronal function, have also been used to analyze PD model flies.

The accumulation of misfolded and/or aggregated αSyn can be assessed by immunohistochemistry, and immunoblotting using detergent (e.g., Triton X-100, sarkosyl, etc.)-separated samples as in other model systems. In addition to the solubility against the detergent, the αSyn with a pathological conformation can be distinguished biochemically by their susceptibility to proteinase K (PK) digestion [34]. PK-resistant αSyn was reported to be accumulated in the brains of PD patients and animal models of PD, including mice and *Drosophila* [35,36,37]. Prasad and colleagues developed an in vivo assay to monitor αSyn oligomerization using the biomolecular fluorescence complementation assay, and found that alterations in cellular protein degradation systems strongly affected oligomeric αSyn accumulation [38].

Compared with *Drosophila* models of other neurodegenerative diseases, assessing the toxicity of αSyn takes a relatively long time (20–30 days after eclosion). To solve this problem, several methods have been used to increase the expression levels of the αSyn protein, which mimics familial PD-linked multiplication mutations that increase the expression levels of αSyn [12,13] (Table 3). Adding the *Drosophila* Kozak sequence CAAA [39] before the start codon was shown to increase the expression of αSyn by two- to five-fold compared with previously reported transgenic lines [40]. However, DA neuron loss was not detected earlier than in the previous models, even though flies bearing two copies of the *TH*-GAL4 driver and *UAS*-WT αSyn transgenes were used. In contrast, codon optimization of the αSyn gene for *Drosophila* to enable more efficient protein translation resulted in an approximately 20-fold increase in αSyn expression compared with the non-codon optimized construct [32]. Flies expressing codon-optimized WT αSyn under the control of the *Rh1*-GAL4 driver (active in the photoreceptors R1–R6) demonstrated ERG abnormalities even at one day after eclosion, and this abnormality progressed with age. This codon-optimized fly line also demonstrated robust retinal tissue degeneration on day 10, and this phenotype was utilized for the biological validation of candidate PD-associated genes identified from large-scale, whole-exome sequencing [41]. Feany’s group used a recently established binary expression system, the Q system, which relies on transcriptional activation by the *Neurospora* protein QF2 to activate the expression of the transgene downstream of the *QF upstream activating sequence* (*QUAS*) [42]. They observed higher levels of αSyn expression than with the conventional GAL4/*UAS* system. The levels of αSyn in the fly brains were found to be almost equivalent to that of human brain homogenates. These model flies showed robust neurodegeneration, locomotor dysfunction, and αSyn aggregation around 10 days after eclosion.

## 3. Familial PD-Causing Mutations of αSyn

Since the discovery of the first missense mutation A53T in αSyn in 1997, five more pathogenic point mutations (A30P, E46K, H50Q, G51D, and A53E), as well as duplication and triplication mutations in αSyn, have been identified as causes of dominantly inherited PD [5,6,7,8,9,10,11,12,13]. In the early 2000s, four groups independently established WT, A30P, and A53T αSyn-expressing fly lines using GAL4/*UAS* systems [18,43,44,45] (Table 4). Histological analyses of DA neurons of flies expressing αSyn under the 3,4-dihydroxyphenylalanine-l-decarboxylase (*ddc*)-GAL4 driver were commonly performed by three groups. The Feany group and Bonini group showed that DA neuron loss was observed in the flies 20 to 30 days after eclosion [18,43], whereas the Mardon group did not observe DA neuron loss even at 30 days in their own fly lines as well as those of Feany and Bonini [45]. Even using other drivers, i.e., *embryonic lethal abnormal vision* (*elav*)-GAL4 and the *glass multiple reporter* (*GMR*)-GAL4, they also failed to detect the αSyn toxicity in the locomotor function and the compound eye morphology.

Considering the importance of αSyn in the pathogenesis of PD, elucidating the pathomechanisms by which αSyn mutations gain neurotoxicity is crucial for understanding PD pathogenesis. In the two early studies performed by the Feany and Bonini groups, greater toxicity for A30P and A53T than WT was commonly observed, despite using completely different experimental systems [18,43]. In addition, Ser129 phosphorylation of αSyn was most abundant in the A53T mutant, followed by A30P and WT in the strains established by the Iwatsubo group [44].

As the E46K, H50Q, G51D, and A53E mutations of αSyn were found later than the A30P and A53T mutations [7,8,9,10,11], αSyn-expressing flies bearing these mutations were reported only recently. In vitro studies showed that A53T, E46K, and H50Q accelerate the rate of αSyn fibril formation, whereas A30P, G51D, and A53E delay it [49,50,51,52]. In contrast, enhanced oligomer formation is considered to be a shared property of the A30P and A53T mutants, which explains their association with early onset PD [49]. However, this is not consistent with the G51D and A53E mutations, which showed overall slow aggregation in vitro [10,52]. Mohite and colleagues established flies expressing each of the E46K, H50Q, and G51D αSyn mutations [46]. They found that E46K, H50Q, and G51D αSyn-expressing flies showed more severe declines in locomotor function than WT αSyn-expressing flies, and the enhanced toxicities of these αSyn mutations strongly correlated with the enhanced production of oligomers. On the other hand, our group established flies expressing WT αSyn and the A30P, E46K, H50Q, G51D, and A53T mutants, using site-specific transgenesis, by which the transgene is inserted into the same locus of the genome, and, thus, the transgenes are expected to be expressed at equivalent levels [47]. The neuronal expression of E46K, H50Q, G51D, and A53T αSyn in flies resulted in stronger toxic effects than the expression of WT αSyn. We also found that the protein level of only E46K αSyn was higher than the other αSyn proteins, despite their equivalent mRNA levels, and demonstrated that the degradation of the E46K αSyn protein was significantly slower than WT αSyn. These results suggest that one of the effects of the E46K mutation in PD pathogenesis is conferring resistance to degradation. More recently, another E46K αSyn-expressing fly line was established using random transgenesis [48]. The authors compared fly lines expressing equivalent levels of WT and E46K αSyn protein. Their results indicated that E46K was more toxic than WT αSyn regarding compound eye degeneration, DA neuron loss, and lifespan, although statistical analyses were not shown. These results show that a shared mechanism causing enhanced toxicity that applies to all αSyn mutants still remains elusive, and each mutation may have different and multiple mechanisms for their toxic effects.

## 4. Genetic Interactions of αSyn with Familial PD-causing Genes

Genetic research on PD has led to the identification of causative gene mutations responsible for familial PD, as well as genetic risk gene variants associated with sporadic PD. Monogenic forms, caused by mutations in a single dominantly or recessively inherited gene, are relatively rare types of PD, accounting for about 5% to 10% of all PD cases. In this section, we focus on *Drosophila* studies on the genetic interaction of αSyn with other familial PD-causing genes, because αSyn is considered to play a crucial role in the pathogenesis of PD, although its accumulation is absent in some familial forms of PD [53].

Loss-of-function gene mutations of *Parkin*, which encodes Parkin E3 ubiquitin ligase, is responsible for an autosomal recessive form of PD (PARK2), in which patients develop typical Parkinsonian symptoms as a result of midbrain DA neuron loss, but usually lack LBs [54]. Several studies have shown that the coexpression of Parkin rescues αSyn-induced DA neurodegeneration, retinal degeneration, and motor dysfunction in WT and A30P αSyn flies [55,56]. Mutations in the *PTEN-induced putative kinase 1* (*PINK1*) gene cause another form of autosomal recessive PD (PARK6) [57]. PINK1 and Parkin work in conjunction to promote mitochondrial quality control; PINK1-dependent phosphorylation of Parkin mediates the clearance of damaged mitochondria, i.e., mitophagy [58]. Similar to Parkin, PINK1 overexpression suppresses αSyn-induced phenotypes in *Drosophila*, including the loss of climbing ability, shortened lifespan, neurodegeneration, and mitochondrial fragmentation [59,60,61]. These studies suggest the protective role of Parkin and PINK1 in αSyn toxicity, but whether a direct molecular interaction between Parkin/PINK1 and αSyn contributes to these effects remains unclear.

Studies of another autosomal recessive PD-causing gene, *phospholipase A2 G6* (*PLA2G6*), have suggested the role of lipid metabolism in the pathogenesis of PD. PLA2G6 encodes iPLA2β, a Ca^2+^-independent phospholipase A2. Recessive mutations in PLA2G6 cause infantile neuroaxonal dystrophy and atypical neuroaxonal dystrophy [62,63], as well as PLA2G6-related dystonia-parkinsonism, also called Parkinson disease 14 (PARK14) [64,65]. Lin and colleagues reported that the loss of iPLA2-VIA, the fly homolog of PLA2G6, shortens lifespan, impairs synaptic transmission, and causes neurodegeneration through an increase in ceramide levels [66]. Interestingly, the overexpression of αSyn also increases ceramide levels, and αSyn-induced retinal degeneration was suppressed by myriocin, a potent inhibitor of de novo ceramide synthesis. In contrast to this report suggesting the role of lipid metabolism as a downstream effector of αSyn toxicity, another recent report suggested the role of lipid metabolism as an upstream effector of αSyn aggregation. Mori and colleagues demonstrated that iPLA2-VIA regulates αSyn stability through membrane remodeling [33]. iPLA2-VIA deficiency induced a shortening of the phospholipid acyl-chain length, which decreases αSyn affinity to the synaptic membrane, leading to αSyn aggregation.

Leucine rich-repeat kinase 2 (LRRK2) is a member of the ROCO family of G-proteins. In 2002, linkage analysis identified a novel locus, *PARK8*, in a large Japanese family with multiple affected generations [67]. Subsequently, two independent groups identified mutations in the *LRRK2* gene as causes of an autosomal dominant form of PD [68,69]. LRRK2 consists of multiple domains, including armadillo repeats, ankyrin repeats, leucine-rich repeats, Ras of complex, C-terminal of Roc, kinase, and WD40 domains, suggesting that LRRK2 has diverse binding partners [70]. The *Drosophila* homolog of LRRK2, Lrrk, is suggested to have a protective function, because the loss-of-function *Lrrk* mutant flies show a shortened lifespan and locomotor dysfunction [23,71]. In contrast, it is reported that a knockdown of *Lrrk* suppressed heat-induced paralysis caused by the loss of the mitochondrial chaperone *Hsc70-5* [72], suggesting that the role of Lrrk depends on the situation. The overexpression of familial PD-linked LRRK2 I1915T mutant results in DA neuron degeneration and a shortening of lifespan [73]. By using this model, Sun and colleagues demonstrated that the down-regulation of the chromatin-remodeling factor SMRCA4, which was found by a gene co-expression analysis on human PD patient microarray datasets, prevents the phenotypes of the LRRK2 I1915T-expressing flies [73]. It is also reported that the overexpression of LRRK2 G2019S induces the synaptic autophagy in the neuromuscular junction, although the role of synaptic autophagy induction was not addressed [74]. Regarding the relationship between Lrrk and αSyn, it is suggested that the abnormal regulation of the actin cytoskeleton, and downstream mitochondrial dysfunction, are convergent mediators of αSyn- and LRRK2-associated neurotoxicity [75]. The authors found that monomers and dimers of Lrrk promote normal actin dynamics [76]. In contrast, the oligomerization of Lrrk, which is accelerated by PD-linked mutations, promotes the stabilization of F-actin and also enhances αSyn neurotoxicity [75]. Interestingly, a clinically protective mutant reduced Lrrk oligomerization and αSyn neurotoxicity, by which αSyn causes the disruption of spectrin-mediated F-actin dynamics [42]. These studies provide a specific mechanistic link between αSyn and LRRK2.

The autosomal dominant PD-causing genes *vacuolar protein sorting 35* (*VPS35*) and *DnaJ heat shock protein family member C13* (*DNAJC13*) are both associated with intracellular membrane trafficking. VPS35 is a vital element of the retromer complex and mediates the retrograde transport of cargo from the endosome to the trans-Golgi network [77]. A missense mutation (p.D620N) in the *VPS35* gene was identified as the cause of an autosomal dominant form of PD (PARK17) [78,79]. In cultured cells, VPS35 knockdown perturbed the maturation step of cathepsin D in parallel with the accumulation of αSyn in lysosomes [80]. Consistent with these results, VPS35 knockdown induced the accumulation of the detergent-insoluble αSyn species in the brain and exacerbated both locomotor dysfunction and compound eye degeneration in flies expressing αSyn [80]. DNAJC13 is an endosome-associated protein that is thought to regulate endosomal membrane trafficking. The *DNAJC13* gene has been identified as a causative gene for the autosomal dominant familial form of PD (PARK21) [81,82]. Yoshida and colleagues demonstrated that PD-linked N855S mutant DNAJC13 causes αSyn accumulation in the endosomal compartment, presumably owing to defective cargo trafficking from early endosomes to late and/or recycling endosomes [83]. In vivo experiments using αSyn transgenic flies showed that mutant DNAJC13 not only increased the amount of insoluble αSyn in fly heads, but also induced DA neuron degeneration, the rough eye phenotype, and age-dependent locomotor impairment.

Heterozygous mutations in the *Coiled-coil-helix-coiled-coil-helix domain-containing protein 2* (*CHCHD2*) gene were identified by linkage analysis and whole genome sequencing of Japanese families with an autosomal dominant form of PD (PARK22) [84]. Several studies identified *CHCHD2* variants that are associated with PD [85,86,87,88] and dementia with Lewy bodies (DLB) [88], whereas other studies did not find evidence that *CHCHD2* is linked to PD [89,90,91]. The CHCHD2 protein localizes to the intermembrane space of mitochondria, and is suggested to play a role in the regulation of mitochondrial respiration, transcriptional regulation of complex IV, and mitochondria-associated apoptosis [84,92,93]. The loss of CHCHD2 in *Drosophila* causes abnormal matrix structures and impairs oxygen respiration in mitochondria, leading to oxidative stress, DA neuron loss, and motor dysfunction with age [92]. A neuropathological study on an autopsy case harboring the CHCHD2 T61I mutation demonstrated widespread αSyn pathology, and *Drosophila* expressing CHCHD2 T61I in DA neurons had accelerated αSyn aggregation, which was also reproduced in DA neuron cultures from CHCHD2 T61I-induced pluripotent stem cells [94].

## 5. Genetic Interaction of αSyn with Risk Genes for Sporadic PD

Monogenic forms of PD represent less than 10% of PD in most populations, whereas the vast majority of PD is considered to result from complex interactions between multiple genetic and environmental factors [95]. It is important to identify the factors that contribute to the development of sporadic PD, to establish prevention and treatment measures that are applicable to a large number of patients.

High temperature requirement protein A2 (HtrA2, also known as Omi) is a homolog of the bacterial heat shock protein HtrA [96] and has been demonstrated to be a susceptibility locus for PD (PARK13). Two mutations (S141A and G399S) that are adjacent to two putative phosphorylation sites were found in German PD patients [97]. In addition, genetic variability in the *HtrA2* gene was subsequently reported to contribute to the risk of PD in different populations worldwide [98,99]. Chung and colleagues showed that HtrA2 specifically recognizes and degrades oligomeric αSyn, but not monomeric αSyn, in vitro [100]. In vivo experiments using transgenic flies and mice also supported these notions. The coexpression of human HtrA2 prevented the formation of αSyn aggregates and neurodegeneration in a fly model. *Drosophila* HtrA2/Omi also exerts protective functions against αSyn-induced neurotoxicity [101].

The *β-glucocerebrosidase 1* (*GBA1*) gene encodes the lysosomal hydrolase glucocerebrosidase (GCase); recessive mutations in *GBA1* cause Gaucher’s disease (GD) [102], and it is now known to be the strongest genetic risk factor for sporadic PD. GCase is an enzyme involved in sphingolipid metabolism, catalyzing the conversion of glycosphingolipids into glucose and ceramide, and its absence in GD leads to the accumulation of glucosylceramide (GlcCer) in macrophages and neuronal cells. Multicenter genetic analyses demonstrated that heterozygous GD-associated mutations in *GBA1* genes increase the risk of PD (OR 5.43) [103], and also that of DLB (OR 8.28), in which αSyn inclusions are abundantly found in the brain [104]. Hypotheses proposed to explain this association include a gain-of-function owing to mutant GCase, and enzymatic loss-of-function, both of which promote αSyn aggregation. The finding that most mutant *GBA1* alleles result in a misfolded GCase protein supports a gain-of-function role for mutations in *GBA1*. Maor and colleagues showed that transgenic flies carrying mutant human N370S, L444P, and 84GG variants demonstrated an activation of the unfolded protein response and developed PD-like phenotypes, such as the loss of DA neurons, locomotor defects, and a shorter lifespan [105]. In addition, coexpression of mutant GCase with αSyn in DA neurons delays αSyn degradation, leading to αSyn aggregation [106]. On the other hand, a loss-of-function mechanism is supported by the fact that some *GBA1* null mutations, such as 84GG and IVS2 + 1, have been reported in patients with PD [107]. Moreover, carriers of severe GBA1 mutations (84GG, IVS + 1, V394L, D490H, L444P, and RecTL) have a much higher risk of developing PD than those with mild mutations (N370S and R496H) [107]. Our group has analyzed the effects of GCase deficiency on the neurotoxicity of αSyn in a fly model [108]. GCase deficiency caused by *GBA1* gene knockdown in flies induced the accumulation of the CGase substrate GlcCer and PK-resistant αSyn in the fly brain, and aggravated locomotor dysfunction and retinal degeneration. In vitro experiments demonstrated that GlcCer directly promotes the conversion of recombinant αSyn into the PK-resistant form. Abul Khair and colleagues also showed that GCase deficiency increased the levels of Triton-X100-insoluble αSyn, and aggravated DA neuronal loss, locomotor defects, and abnormal sleep behavior induced by WT, A30P, and A53T αSyn [109]. Davis and colleagues showed the mild enhancement of αSyn toxicity and αSyn aggregation by GCase deficiency, but they concluded that the deleterious consequences of mutations in *GBA1* are largely independent of αSyn [110].

*Arylsulfatase A* (*ARSA*) is a gene responsible for metachromatic leukodystrophy, an autosomal recessive lysosomal storage disorder. Recently, pathogenic and protective mutations in *ARSA* have been found to be linked to PD [111]. ARSA deficiency was shown to increase the aggregation, secretion, and propagation of αSyn in cells and nematodes. Moreover, ARSA was found to directly interact with αSyn in the cytosol, and the interaction was more extensive for protective ARSA variants, and less so for pathogenic ARSA variants than WT ARSA. Consistently, the ectopic expression of ARSA reversed the αSyn-induced locomotor dysfunction in a fly model. However, large international consortiums have not been able to confirm a significant association between *ARSA* and PD [112]. Therefore, additional large-scale studies are necessary to determine whether *ARSA* is associated with PD.

In the previous 10 years, many studies have been conducted based on the ‘common disease-common variant hypothesis’, which assumes that the genetic risk factors for common diseases, such as sporadic PD, are attributed to common variants with a high frequency (>5%) in the population [113]. In 2009, a GWAS of PD identified polymorphisms in *SNCA*, *LRRK2*, and *PARK16* (*NUCS1-RAB7L1-SLC41A1*), *microtubule-associated protein Tau* (*MAPT*), and *bone marrow stromal cell antigen 1* (*BST1*) as common genetic risk factors for PD [15,16]. *MAPT* and *BST1* were identified as loci showing population differences in these studies. Since then, numerous GWAS with increasing numbers of participants have been performed across populations. The most recent and largest PD GWAS to date has identified 90 independent risk variants associated with sporadic PD [114].

Tau, a product of *MAPT*, is a highly soluble microtubule-associated protein, and has been linked to tauopathies, including Alzheimer’s disease [115]. The involvement of Tau in PD has been implicated in pathological studies in which Tau-immunoreactive LBs were detected in the brains of sporadic PD and DLB patients [116,117]. Biochemical analyses also strengthened this link; Giasson and coworkers reported that αSyn induces the fibrillization of Tau, and that the coincubation of Tau and αSyn synergistically promotes the fibrillization of both proteins [118]. Roy and colleagues demonstrated that the coexpression of Tau and αSyn in flies enhanced the rough eye phenotype, as well as the loss of DA neurons compared with either Tau or αSyn expression alone [119]. They also showed that interactions between αSyn and Tau at the cellular level cause a disruption of cytoskeletal organization, axonal transport defects, and aberrant synaptic organization, which contribute to neuronal dysfunction and neuronal death. Interestingly, the presence of Tau led to an increase in urea-soluble αSyn, whereas αSyn did not alter the levels of phosphorylated Tau.

Recently, a unique study combining the advantages of *Drosophila* and GWAS was reported [120]. The authors focused on the fact that the majority of PD risk genes identified in GWAS are expressed in glia at either similar or greater levels than in neurons. To explore the role of individual PD risk genes in glia, they developed a *Drosophila* model of PD in which they can manipulate αSyn and risk gene expression in neurons and glia separately: αSyn is expressed in neurons by using Q-system, and the knockdown of 14 candidate risk genes was induced in glia by using the GAL4/*UAS* system. As a result, *auxilin*, *Lrrk*, *Ras-related protein interacting with calmodulin* (*Ric*), and *Vacuolar protein sorting 13* (*Vps13*), orthologs of the human *Cyclin-G-associated kinase* (*GAK*), *LRRK2*, *Ras like without CAAX 2* (*RIT2*), and *VPS13C* were identified as glial risk factors that modify neuronal αSyn toxicity [120]. The knockdown of each gene increased αSyn oligomerization, suggesting that glia can affect neuronal αSyn proteostasis in a non-cell-autonomous manner.

## 6. Nongenetic Factors of PD

Despite the fact that 90 gene variants have been identified as PD risk factors, it was estimated that these variants only explain 16% to 36% of the heritable risk of PD by prevalence, indicating that many yet-unidentified genetic factors and environmental factors contribute to PD risk [114]. A twin study comparing the concordance rates of PD in monozygotic and dizygotic twins were 15.5% and 11.1%, respectively, suggesting the important role of environmental factors [121]. Age is the biggest risk factor for PD [122,123,124]. In 2012, Noyce and colleagues conducted a meta-analysis of environmental factors for PD; pesticide exposure, well water consumption, and head injury, as well as premotor symptoms, including constipation and depression, have been associated with an increased risk of PD, whereas other factors, such as tobacco, coffee, and alcohol usage have shown possible protective associations with PD [125].

The effects of several environmental factors that have been suggested by epidemiological studies were studied on αSyn-expressing PD flies. Rotenone is known to interfere with the electron transport chain in mitochondria, and is used as a pesticide. The lifetime use of rotenone is known to be associated with PD [126]. When A53T αSyn-expressing larvae were chronically exposed to rotenone, they showed a more severe age-dependent decline in locomotion accompanied by the loss of DA neurons than untreated larvae [127]. As mentioned above, epidemiological studies have demonstrated a significantly reduced risk of PD among coffee and tobacco users [125]. Trinh and colleagues reported that coffee and tobacco, but not caffeine or nicotine, are neuroprotective in PD model flies expressing αSyn [128]. They further demonstrated that the neuroprotective effects of decaffeinated coffee and nicotine-free tobacco require the cytoprotective transcription factor Nuclear factor erythroid 2-related factor 2 (Nrf2) and that a known Nrf2 activator in coffee, cafestol, is also able to confer neuroprotection in fly models of PD.

## 7. Concluding Remarks

In this review, we introduced the αSyn-expressing PD model flies that have been established to date, and summarized recent studies that have clarified the molecular pathogenesis of PD. While rodent models generally attract more attention because of their high conservation of genes and neuronal circuits with humans, genetic modeling of PD in rodents has faced some difficulties [129]. It is important to realize that it may be difficult to fully recapitulate the key pathological and clinical features of human PD in a single model system. Hence, combinatorial studies using different models may provide further insights into the pathogenesis of PD.

Recently, a growing body of evidence has suggested that misfolded αSyn has prion-like properties, in which the native form of αSyn is converted into misfolded forms, and are transmitted from cell to cell, leading to its spread throughout the brain [130]. In recent years, the injection of αSyn preformed fibrils into the brain parenchyma of rodents has been applied for modeling the propagation of αSyn pathology in PD [131]. Although applying such injection experiments in *Drosophila* is difficult because of its small brain size, genetically induced fly models of αSyn transmission are expected to be established in the future, like as in other neurodegenerative disease models [132,133,134,135].

With the development of genetic analysis technology, numerous susceptible genetic factors for PD are being identified, but the biological significance of these genetic factors have largely been left unstudied to date. The combinatorial modelling of multiple genetic factors is required to recapitulate the features of PD because their individual contribution to pathogenesis is relatively small. Considering the convenience of the fly in creating libraries of multiple genetic models, *Drosophila* will continue to provide important insights into the pathogenesis of PD.

## Figures and Tables

**Table 1 ijms-23-01519-t001:** Advantages of using *Drosophila* for studying human diseases.

Advantage	Note
(1) Analysis of gene functions in vivo	Encode homologues of more than 75% of human disease genes
(2) Rapid generation cycle and short lifespan	10–14 days from embryo to adult
	Lifespan of 60–80 days
(3) Well-maintained information	Flybase ^1^, the leading database for genomic and genetic information of *Drosophila*, is linked to a wide range of other available tools
(4) Abundant useful tools for genetic analysis	Genome-wide mutant and RNAi fly libraries
	Cell-type- and temporal-specific gene expression systems
	Genome editing systems
(5) Little labor and cost-effective	Transgenic flies can be established relatively easily at low cost
	Mutant, RNAi, and transgenic flies are available from public stock centers at low cost
	Small space is required for their maintenance

^1^ http://flybase.org (accessed on 23 January 2022).

**Table 2 ijms-23-01519-t002:** Assays to evaluate PD-associated phenotypes in *Drosophila*.

Category of Phenotype	Specific Phenotype Evaluated	Assay
Behavior	Locomotor dysfunction	Climbing assay, automated tracking systems
	Abnormal sleep behavior, circadian rhythm	*Drosophila* Activity Monitor system
	Olfactory deficits	Odor acuity/discrimination assay
	Anxiety	Open-field assay
	Cognitive dysfunction	T-maze assay
	Lifespan	Lifespan assay
Neurodegeneration	DA neuron loss	Counting DA neurons either by tyrosine hydroxylase staining or reporter expression
	Compound eye degeneration	Observation of external eye appearance by light microscopy, scanning electron microscopy, or analysis of retina tissue sections
Neuronal dysfunction	Electrical activity of the retina	Electroretinography
	Electrical activity of brain or motor neurons	Electrophysiological recordings from projection neurons or neuromuscular junction
αSyn accumulation/inclusion formation	αSyn inclusions	Immunohistochemistry with an αSyn antibody
	αSyn aggregation	Immunoblotting of lysates separated by detergent
	Pathological αSyn conformers	Immunohistochemistry with an αSyn antibody after proteinase K (PK) treatment
		Immunoblotting of lysates treated with PK
	αSyn oligomers	Biomolecular fluorescence complementation assay

**Table 3 ijms-23-01519-t003:** αSyn-expressing fly lines with modifications for increasing αSyn expression levels.

Reference (Corresponding Author)	Modifications	Driver Line ^1^	Behavior /Neuronal Function	Histology /Biochemistry	Notes
L.J. Pallanck [40]	Added Kozak sequence (CAAA) and used strains bearing 2 copies of transgenes	*TH*-GAL4		DA neurons at PPL1 ↓ (20 days)	2–5-fold higher αSyn protein level than previously reported lines (Feany [18], Bonini [43])
J.M. Shulman [32]	Codon optimization for *Drosophila*	*Rh1*-GAL4	Progressive ERG abnormalities (1–30 days)	Retina and photoreceptor degeneration (10–30 days)	20-fold increase in αSyn protein level than non-codon optimized line
M.B. Feany [42]	Q system	*Syb*-QF2	Locomotor dysfunction (>7 days)	Brain vacuolization, cortical neuron ↓ (>10 days) Inclusion + (>1 days)	Using the Q-system yielded higher levels of αSyn than using the GAL4/*UAS* system

^1^ Tissues in which each GAL4 or QF2 driver induces *UAS*- or *QUAS*-linked transgene expression are as follows: *TH*-GAL4, DA neurons; *Rh1*-GAL4, R1-6 photoreceptor cells; *Syb*-QF2, pan-neurons. PPL1, protocerebral posterior lateral.

**Table 4 ijms-23-01519-t004:** WT or familial mutant αSyn-expressing fly lines and their characterizations.

Reference (Corresponding Author)	*SNCA* Variant	Driver Line ^1^	Behavior /Neuronal Function	Histology /Biochemistry	Notes
M.B. Feany [18]	WT A30P A53T	*ddc*-GAL4		DA neuron ↓ (30–60 days) Inclusions (1 day)	
		*elav*-GAL4	Locomotor dysfunction (A30P > A53T, WT, >23 days)	DA neuron ↓ (30–60 days) Inclusions (20–30 days) αSyn neuritic pathology (60 days)	
		*GMR*-GAL4		Retinal degeneration (10–30 days)	
N.M. Bonini [43]	WT A30P A53T	*ddc*-GAL4		DA neuron ↓ (A30P > A53T > WT, 20 days) Inclusions + (ubiquitin +) αSyn neuritic pathology (20 days)	
T. Iwatsubo [44]	WT A30P A53T	*elav*-GAL4		Phosphorylation of αSyn at S129 (A53T > A30P > WT)	
G. Mardon [45]	WT A30P A53T	*ddc*-GAL4		No changes in DA neuron number (30 days)	No changes in DA neuron number were found in the Feany [18] and Bonini [43] lines
		*elav*-GAL4	No changes in locomotor function (up to 38 days)		
		*GMR*-GAL4		No changes in ommatidial morphology (40 days)	
S.K. Maji [46]	WT E46K H50Q G51D	*elav*-GAL4	Locomotor dysfunction (G51D > E46K > WT, H50Q, >30 days) Shortened lifespan (G51D, H50Q, E46K > WT)	DA neuron ↓ (30 days) αSyn oligomers (H50Q, G51D, E46K > WT, >10 days)	
Y. Nagai [47]	WT A30P E46K H50Q G51D A53T	*GMR*-GAL4		Mild eye degeneration (1 day) E46K αSyn accumulation (1 day)	Site-specific transgenesis to express equivalent transcriptional levels of αSyn
		*nSyb*-GAL4	Locomotor dysfunction (E46K, H50Q, H50Q, and A53T at 3 weeks, all lines at 5 weeks)	E46K αSyn protein accumulation (1 day)	
M. Haddadi [48]	WT E46K	*GMR*-GAL4		Irregular organization of ommatidia, loss of bristles (10 days) Retinal neuron degeneration	
		*ddc*-GAL4	Locomotor dysfunction (>20 days) Paraquat sensitivity↑	DA neuron ↓ (E46K > WT)	
		*elav*-GAL4	Short lifespan (E46K) Ethanol sensitivity↑		

^1^ Tissues in which each GAL4 driver induces *UAS*-linked transgene expression are as follows: *elav*-GAL4, pan-neurons; *ddc*-GAL4, DAand serotonergic neurons; *GMR*-GAL4, compound eye; *nSyb*-GAL4, pan-neurons.

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
