# Peer review of "Roles of α-Synuclein and Disease-Associated Factors in Drosophila Models of Parkinson’s Disease"

_ijms, 2022, doi:10.3390/ijms23031519_

Round 1

Reviewer 1 Report

It is a nice review with contrasting information that underpins the involvement of α-Syn in the pathophysiology of Parkinson's disease (PD), as well as it places Drosophila melanogaster as a suitable model to study it. It is well-written and easy-to-follow. However, I would appreciate to clarify what are disease-associated factors. I guess that the scope of this review is to discuss the animal model and highlight the importance of α-Syn and other genes in the development of PD. In consequence, I recommend the authors to rewrite the final sentence of the abstract, concerning the goal of the review, as if the disease-associated factors are the genes/proteins implied in the development of PD, not all the described are left uncovered.

Reviewer 2 Report

In this review, the authors focused on the Drosophila melanogaster as an experimental animal model, specifically, αSyn-expressing PD model flies, to highlight its contributions in understanding the molecular pathogenesis of PD (o neurobiological mechanisms underlying pathogenesis). Methods and experimental approaches related to this animal model are exhaustively illustrated and well discussed by authors. The manuscript is well organized, the different sections are structured and clearly written, and the literature cited is appropriate to support the evidence discussed, but the AA  should update the relevant literature about the model Lrrk.
